# Regressing Image Sub-Population Distributions with Deep Learning

**DOI:** 10.3390/s22239218

**Published:** 2022-11-27

**Authors:** Magdeleine Airiau, Adrien Chan-Hon-Tong, Robin W. Devillers, Guy Le Besnerais

**Affiliations:** 1ONERA, Université Paris Saclay, 91123 Paris, France; 2CNES, 75039 Paris, France

**Keywords:** deep learning, fairness, image classification, objects detection in image

## Abstract

Regressing the distribution of different sub-populations from a batch of images with learning algorithms is not a trivial task, as models tend to make errors that are unequally distributed across the different sub-populations. Obviously, the baseline is forming a *histogram* from the batch after having characterized each image independently. However, we show that this approach can be strongly improved by making the model aware of the ultimate task thanks to a *density loss* for both sub-populations related to classes (on three public datasets of image classification) and sub-populations related to size (on two public datasets of object detection in image). For example, class distribution was improved two-fold on the EUROSAT dataset and size distribution was improved by 10% on the PASCAL VOC dataset with both RESNET and VGG backbones. The code is released in the GitHub archive at achanhon/AdversarialModel/tree/master/proportion.

## 1. Introduction

Image classification performance has risen thanks to deep learning approaches [1,2]. Yet, data driven algorithms are still far from being perfect, in particular when testing data are sampled from a different distribution than the training data (also called dataset drift), e.g., out-of-distribution detection [3], transfer learning [4], domain adaptation [5], and adversarial attacks [6].

In addition, those deep learning models tend to be unfair toward some sub-populations of the distribution, as highlighted in numerous papers in the literature (see [7] for a review). Of course, a detector with no error is fair. Thus, in critical use cases such as autonomous driving, fairness is not really an issue because the models should have an almost perfect detection rate for all sub-populations of pedestrians regardless (independently from color of skin, age, size, gait, clothes, presence of medical walker, etc.).

Now, in less critical use cases, fairness should be achieved before perfection. A detector which misses 50% of sub-population 1 and 50% of sub-population 2 will still perfectly regress the distribution, while improving the model by removing errors on sub-population 1 will increase the gap between the predicted and true distributions.

This paper specifically deals with the estimation of the distribution of different sub-populations into a batch of data. We consider two types of sub-populations based on classes and sizes. For the classes, given a batch of data, one wants to know the proportions of the classes within the batch. Those proportions could asymptotically meet the global statistic. Yet, one may remark that standard classification has exactly this problem when the batch size is one.

More formally, we have an underlying multi-class classification problem given by a probability density *P* on a data space *X* and a ground truth function *y* from *X* to the label space {1,…,Y}. However, in this paper, the task is to regress the (true) distribution:(1)$$\mathrm{true}\_\mathrm{distribution}\left(x_{1},\ldots ,x_{K}\right)=\frac{1}{K}\left(\begin{array}{c} \left|\{k,y\left(x_{k}\right)=1\}\right|\\ \ldots \\ \left|\{k,y\left(x_{k}\right)=Y\}\right| \end{array}\right)$$
for any given batch $x_{1},...,x_{K}$ sampled from *P* (|.| is the cardinal of the set).

For the sizes, one way to see the problem is that one has a detector, and, given a batch of data, one wants to know the size distribution of an object of interest within the batch (more detail is given later—see Equation (Equation 3)).

Despite its appearance, this batch regression problem has real applications. For example, some illnesses [8] can be detected by an abnormal ratio of cell sub-populations independent of the total number of cells. Furthermore, for traffic monitoring, it can be relevant to focus on the distribution of the different kind of vehicles (independently from the absolute number of each) either for road traffic [9] or marine traffic [10]. Finally, in agriculture, there are situations where the distribution of sizes of fruits should be monitored to select the harvesting time [11]. This problem of size distribution also arises in materials for improving alloy quality [12]. Another use case is the estimation of aluminum particle size distribution in solid propellant during combustion which cannot be deduced from the initial size distribution as combustion leads to fragmentation and/or amalgamation [13,14].

Importantly, this problem of sub-population estimation is classical from a statistical point of view. However, in this paper we consider this problem when data are images. In this case, we need to rely on deep networks to deal with the high dimensionality of the images. Thus, tackling this problem by modeling the batch distribution itself can be inefficient, as it increases the dimensionality of the data. Thus, relying on the intermediary elementary data classification and then post-processing the predictions to extract a distribution at the batch level may be a better framework.

The contribution of this paper is the benchmarking of multiple ways to implement this framework: naively, with data selection, with normalization, etc. In addition, this paper shows that making the classifier aware of the ultimate task by managing to compute a gradient through distribution comparison is better than four other methods inspired by selective classification or fairness tools. The offered pipelines are presented by Figure 1 and Figure 2 for class and size sub-populations, respectively.

Importantly, it is rather obvious that making an algorithm more aware of the ultimate task will improve the ultimate performance. However, here we obtain significant improvement of the density estimation. For example, on the EUROSAT dataset, we halve the class distribution error of the baseline using our density loss, and on the PASCAL VOC dataset, we improve the quality of the size density estimation by 10%.

The amplitude of this improvement shows that this problem should receive more attention from the community, as all methods are clearly not equivalent.

To summarize, the contributions of this paper are:The consideration of batch distribution estimation for class and size sub-populations.The offering of a new loss based on distribution comparison.A demonstration that classifier made aware of the ultimate task thanks to this loss has significantly better predicted distribution than several baselines in many different experiments on classical computer vision datasets (and with different CNN backbones).

## 2. Related Works

As pointed out in the introduction, to the best of our knowledge, distribution estimation from image batches has currently received little attention from the community (despite that this problem has real applications and theoretical relevance). Of course, this problem of sub-populations is not new (see, for example, [15]) but this paper specifically tackles both sub-population and image data simultaneously.

From a theoretical point of view, one could think about solving Equation (Equation 1) by directly training a regressor which takes a batch as input and produces a distribution and comparing this distribution to the ground truth distribution. This can be performed, for example, by flattening the batch using a pooling operation (which will ensure invariance toward permutation within the batch). Such an idea has been introduced in a very different context for point cloud classification [16]. This method is called *simple regression* in this paper, but it is found to be dramatically ineffective in our experiments on image batches. Indeed, the drawback of this method is that direct distribution modeling increases the dimensionality of the data (and decreases the number of samples) so that the training is no longer efficient. Alternatively, the straightforward baseline (called the *baseline* in this paper) for estimating sub-population density is the computation of the distribution after having characterized each sample. More formally, this *baseline* aims to solve Equation (Equation 1) by only training a classifier *f* on the underlying classification problem P, y, hoping that $\left(\begin{array}{c} \left|\{k,f\left(x_{k}\right)=1\}\right|\\ \ldots \\ \left|\{k,f\left(x_{k}\right)=Y\}\right| \end{array}\right)\approx \left(\begin{array}{c} \left|\{k,y\left(x_{k}\right)=1\}\right|\\ \ldots \\ \left|\{k,y\left(x_{k}\right)=Y\}\right| \end{array}\right)$. One may even just consider $\sum_{k}\tilde{f}\left(x_{k}\right)$ with $\tilde{f}$ being the likelihood of *Y* as the predicted distribution. Currently, this *baseline* behaves much better than *simple regression*.

Yet, in this *baseline*, *f* is not aware of the ultimate task, and it is known that a classifier could be unfair between the different classes (e.g., it could even never predict a specific class while still having a $1-\frac{1}{\left|Y\right|}$ accuracy in balanced case). Hence, here, this unfair behavior will damage density prediction.

### 2.1. Fairness and Sub-Population-Based Losses

An issue with the *baseline* method is that the individual classification error may be poorly distributed across the sub-populations, resulting in a bad density estimation even under relatively high accuracy.

Thus, the first aspect of a solution is the penalization of the model more when the distribution of the errors has low entropy at a global level. For example, Dice loss [17] has been introduced to optimize intersection over union (IoU), which penalizes the biased distribution of errors.

However, the main effect of Dice loss is that it gives equal importance to each class independently from the class ratio at the distribution level. Indeed, let consider a classification problem with two classes; if P(y(x)=0)=95%, then, the trivial classifier f(x)=0 already has a 95% accuracy but only a 47% IoU. Inversely, if P(y(x)=0)=50%, then an accuracy of 95% guarantees an IoU of at least 0.81%. Let $\left(\begin{array}{cc} a & b\\ c & d \end{array}\right)$ be the confusion matrix. Then, $\mathrm{Accu}=\frac{a+d}{a+b+c+d}$ and $2\mathrm{IoU}=\frac{a}{a+b+c}+\frac{d}{b+c+d}$. Without loss of generality, one can normalize a+b+c+d=1, leading to Accu=1−(b+c) and $2\mathrm{IoU}=\frac{a}{1-d}+\frac{d}{1-a}$. Then, assuming a+c=b+d=0.5, it holds that max(a,d)≤0.5 and thus min(a,d)≥Accu−0.5. Injecting these inequalities into an IoU definition gives $\mathrm{IoU}\geq \frac{\mathrm{Accu}-0.5}{1.5-\mathrm{Accu}}$. Thus, losses such as Dice loss can help to improve *fairness*, mostly when sub-populations are not balanced at the distribution level, which is not the case in this paper. Still, this *baseline+Dice* method will be considered in the experiments to follow. The other limitation of this method is that it is relevant only when the outputs of *f* directly match the sub-populations (this will not be the case for size-based sub-populations).

The idea of injecting information from the global level meets many fairness approaches based on re-weighting [18,19]. Following [19], we re-weight the outputs of the model to achieve increased fairness by considering global statistics of the predictions averaged over the entire training set. This method is called *baseline+fairness* in this paper. However, this method relies on an ad hoc re-weighting, while the method offered in this paper will basically learn the re-weighting.

### 2.2. Selective Classification

As the goal is the regression of a batch’s proportions, individual classifications are a method but not the ultimate goal. Thus, one could wonder if selective classification [20] (or classification with rejection) can be relevant. Indeed, rejecting samples on which the classifier has low confidence is known to empirically increase the accuracy on the accepted samples (despite this can not be always true from theoretical point of view, in particular if confidence is independent from errors).

However, the situation here is not the same, because accuracy is not the ultimate goal. Thus, the question of whether confidence-based rejection will help proportion estimation is not clear: one can wonder if it is relevant to remove a sample due to strong ambiguity between classes 1 and 2 while simply saying that the fact that the confidence for class 1 is 0.5 and the confidence for class 2 is 0.5 is still information relevant to the other classes (classes 3, 4, etc.). In addition, selective classification has already been suspected to decrease fairness [21]. Furthermore, [22] stresses that small objects tend to receive lower confidence values than larger objects. Thus, if the two sub-populations are small and large objects, the rejection will penalize the prior, worsening the proportion estimation between the two sub-populations.

So, an additional minor contribution of this paper is testing of whether confidence-based rejection helps proportion regression. Formally (in the case of a sub-population of classes), it consists of approximating the $\frac{\left|\{k,y\left(x_{k}\right)=c\}\right|}{K}$ distribution by $\sum_{k,C(\tilde{f}\left(x_{k}\right))\geq \sigma }\tilde{f}\left(x_{k}\right)$ instead of $\sum_{k}\tilde{f}\left(x_{k}\right)$ where $C(\tilde{f}\left(x_{k}\right))$ is a criterion estimating the confidence of *f* on datum $x_{k}$. This algorithm will be called *baseline+rejection* in this paper.

## 3. Proportion-Aware Loss

The previous section describes several baselines for sub-population proportion estimation from image batches. Yet, none of those baselines seems completely satisfying:*simple regression* does not take advantage of the fact that the problem is structured, worsening training by increasing the number of dimensions and reducing the number of samples;*baseline* is not aware of the ultimate task;*baseline+rejection* may or may not improve distribution extracted from accepted data depending on the correlation between rejection and bias in distribution estimation;*baseline+Dice* is most useful in the unbalanced case and limited to a sub-population which matches the outputs of the classifier;*baseline+fairness* is an ad hoc re-weighting based on the statistics of the entire training set, but this re-weighting can be more efficient if learned from the data.

Aiming to bypass those limitations, we offer to combine a loss in the ultimate task (such as in *simple regression*) but at the top of a standard classifier with standard crossentropy loss (i.e., at the top of the *baseline*). Despite the simplicity of this approach (called ***density loss*** in this paper), it consistently improves proportion regression in experiments in the next section.

Yet, before moving to the experiments, some details have to be presented here. First, as our ultimate goal is to improve proportion regression, we need a metric for distributions which is not as standard as the 0/1 error metric in classification. Then, some careful details in the implementation are required to allow the gradient to be correctly propagated for the final loss.

### 3.1. Metric

Classical metrics for regression problem are usually $L_{2}$ and $L_{1}$ norm with $L_{2}$ being more smooth but less sensible to small variation than $L_{1}$. Furthermore, for measuring a gap between the predicted distribution and the real one, a classical metric is KL divergence [23].

So, we chose to consider the sum of those three terms $L_{2}+L_{1}+D_{KL}$ as a metric to measure the distance between the predicted and real distributions in all experiments in this paper. We found that $L_{1}$ tends to be the preponderant metric ($L_{2}$ has almost no influence as values are below 1) and KL divergence decreases more rapidly than $L_{1}$ in our experiments. We filter the KL divergence as we observe that the *pytorch* implementation may suffer from numerical instability (e.g., being negative while it is mathematically impossible).

Now, the evaluation process of our experiments is the sampling of many batches from a testing set of data (of course unseen by the different models during training), and, for each model and batch, we measure the gap between the real distribution and the one produced by the model with the metric introduced above. Yet, contrary to simple classification, we cannot evaluate all possible batches, and thus, evaluation itself (not only the training) has some randomness.

However, we feel that this randomness is not an issue, as we perform both training and testing steps multiple times and report only average performances.

Precisely, in all our experiments, we perform random batch sampling without replacement until each data have been seen 10 times (i.e., we consider all batches from 10 epochs with shuffled data over testing data). Then, we average the distribution gap of all those batches (all 10 epochs) to obtain the performance for one trial. Finally, we perform five trials (training and testing) for each algorithm.

### 3.2. Gradient Flow

#### 3.2.1. Class Sub-Populations

For class sub-populations, we want to minimize the gap between the vector whose component *c* is $\frac{\left|\{k,f\left(x_{k}\right)=c\}\right|}{K}$ (the true distribution) and $\sum_{k}\tilde{f}\left(x_{k}\right)$ (the predicted distribution). Yet, this assumes that $\left|\right|\tilde{f}\left(x_{k}\right){\left|\right|}_{1}=1$ for all *k*. This can be achieved by *softmax*. Yet, *softmax* offers a poorly conditioned gradient. In cross entropy loss, one virtually takes a log after *softmax*. However, here it would have been the raw *softmax* value. This leads to a small gradient in our experiments. Thus, we chose to consider that the predicted distribution is
(2)$$\mathrm{predicted}\_\mathrm{distribution}\left(x_{1},\ldots ,x_{K}\right)=\frac{\sum_{k}relu\left(\tilde{f}\left(x_{k}\right)\right)+softmax\left(\tilde{f}\left(x_{k}\right)\right)}{\left|\right|\sum_{k}relu\left(\tilde{f}\left(x_{k}\right)\right)+softmax\left(\tilde{f}\left(x_{k}\right)\right){\left|\right|}_{1}}$$
where $\tilde{f}$ is the vector of *logit* (not the likelihood). The underlying idea is the allowance of a gradient flow by relying on the raw *logit*. Yet, to ensure that contributions are positive, we consider the *relu* of the *logit* (which is dominant regarding the *softmax*, but we still add the *softmax* to force a gradient in the situation where all *logit* are negative).

#### 3.2.2. Size Sub-Population

For sub-populations related to the size, we consider the following paradigm:Inputs of the models are $\left(x_{1},s_{1},y_{1}\right),\ldots ,\left(x_{K},s_{K},y_{K}\right)$ with $x_{k}$ being a datum (for example, a resized crop from an image), $s_{k}$ being the size of $x_{k}$ (for example the size of the image crop before resizing), and $y_{k}$ is either 0 or 1 depending on whether $x_{k}$ is a datum of interest or not.The real distribution related to this batch is given by the set of sizes of the data of interest ${\left\{s_{k}\right\}}_{\left\{k,y_{k}=1\right\}}$.In practice, we select a kernel density function G [24] and consider that the true distribution is the function
(3)$$\mathrm{true}\_\mathrm{distribution}\left(\left(x_{1},s_{1}\right),\ldots ,\left(x_{K},s_{K}\right)\right)=t\to \frac{\sum_{k,y_{k}=1}G\left(t-s_{k}\right)}{\left|\{k,y_{k}=1\}\right|}$$Finally, the model *f* tries to predict if the datum is an object of interest (or not), i.e., *f* still predicts *y*. However, the ultimate goal is for the predicted distribution given by $t\to \frac{1}{\left|\{k,f\left(x_{k}\right)=1\}\right|}\sum_{k,f(x_{k})=1}G\left(t-s_{k}\right)$ to be close to the real distribution (defined directly above). The distance between the two distributions is computed using sampling of a predefined regular 1D grid.

More precisely, for improving gradient stability across the kernel density estimation, we allow the predicted distribution to be
(4)$$\begin{array}{c} \mathrm{predicted}\\ \mathrm{distribution} \end{array}\left(\left(x_{1},s_{1}\right),\ldots ,\left(x_{K},s_{K}\right)\right)=t\to \frac{\sum_{k,f(x_{k})=1}\tilde{f}\left(x_{k}\right)G\left(t-s_{k}\right)}{\int \sum_{k,f(x_{k})=1}\tilde{f}\left(x_{k}\right)G\left(\tau -s_{k}\right)d\tau }$$
where $\tilde{f}\left(x_{k}\right)\in \left[0,1\right]$ is extracted from *logit* of *f* (we consider the mean of softmax and softplus in our experiments).

Interestingly, it is known that small objects (e.g., included in 32 × 32 bounding boxes) are harder to detect than larger ones. For example, in most detection datasets, small objects are counted separately. Yet, the exact purpose of the pipeline is to improve the predicted distribution by asking the network to perform as well for small objects as for large ones. Potentially, this may be achieved by simply worsening detection for large objects. Yet, it may still improve the predicted distribution. In addition, in the case of size distribution estimation, the intermediary algorithm may process each image after resizing, i.e., by losing information about the related size. This makes it even more important to make the model aware of the ultimate goal.

## 4. Experiments

The two previous sections have introduced five methods (mainly five different losses) which aim to extract the proportion of sub-populations in a batch. This section presents experiments on multiple datasets which evaluate the behavior of all five algorithms on different datasets/backbones.

Indeed, the offered losses are independent from the underlying backbones; thus, we report results for VGG13, VGG16 [25], RESNET34, and RESNET50 [26] backbones. Yet, the results are consistent across these backbones.

These experiments mainly employ moderately sized datasets. However, we think that these are the use cases where our algorithm would be relevant. Indeed, in very large datasets, there are probably methods of filtering the predicted distribution using information from the entire test set.

If not explicitly mentioned, the batch size is 256 in all our experiments (256 images in classification and 256 box proposal in detection).

### 4.1. Experiment with Sub-Populations Based on Classes

#### 4.1.1. Datasets

We chose to rely on standard classification datasets for evaluating the capacity to predict class proportion in batches of data. We relied on the CIFAR [27], SVHN [28], and EUROSAT [29] datasets. All three datasets have native *pytorch* data loaders.

For CIFAR and SVHN, we relied on the standard training/testing split. For EUROSAT, we used the pytorch loader, which is deterministic. Then, images with IDs that are multiples of three were used for testing while the others were used for training (i.e., 66% training and 33% testing).

#### 4.1.2. Implementation Details

For CIFAR, all algorithms were trained for 40 epochs with the ADAM optimizer [30] with a learning rate starting at 0.001 and a decay of 0.1 for the last 20 epochs. Early termination occurred when training accuracy reached 99%.

For selective classification, as we are considering proportion at batch level (and not at distribution level), it is not clear that the rejection mechanism should be implemented using a global statistic, e.g., using absolute thresholds. Indeed, those thresholds can otherwise be inadequate for a given batch. Thus, in the following, we estimate the proportion using only the 80% most confident sample from the batch, with the confidence estimated by the inverse of softmax entropy.

#### 4.1.3. Results

The results are presented in Table 1, Table 2 and Table 3 for the CIFAR, SVNH, and EUROSAT datasets. First, as pointed in the introduction, *simple regression* offers very poor performances: processing the batch directly without taking into account the intermediary image structure seems to lead to the problem of an excessively high number of dimensions.

*Baseline+fairness* is the best on CIFAR but has a poor performance on SVHN and is only average on the EUROSAT dataset. *Baseline+Dice* is inversely the best on EUROSAT but has a poor performance on CIFAR. *Baseline+rejection* improves slightly upon the *baseline* on average. This last result shows that discarding ambiguous samples is not necessarily a good idea when regressing a proportion. Indeed, wrong but overconfident samples are here more harmful than ambiguous ones.

Inversely, *density loss* performs well on all datasets and is consistently either the best or the second-best method for all three datasets and four backbones. Importantly, even when *density loss* is not the best, it is only slightly below the best (*Dice* on EUROSAT, *fairness* on SVHN). This result stresses that *density loss* is the best solution based on the average performance across all datasets. Figure 3 summarizes this claim.

#### 4.1.4. Influence of Batch Size

Batch size is a critical parameter of all these experiments. Indeed, if batch size is one, then the problem collapses into simple classification. Inversely, if the batch size becomes very large, then all proportions tend to $\frac{1}{\left|Y\right|}$ where |Y| is the number of classes, making all the methods converge toward the same value. Thus, the offered algorithm is mostly relevant for moderate batch sizes. However, moderate batch sizes are the ones found in deep learning as excessively small batches lead to time wastage when moving data from the CPU to the GPU and excessively large batches cannot be loaded in the GPU. However, in order to see the sensitivity to batch size, we considered batch sizes of 128, 256, and 512 on EUROSAT for the baseline and our density loss. Those results (which are not averaged over five trials, contrary to the other results from this paper) are summarized in Table 4. We can see that results are consistent across these values, stressing that the experiments presented in this paper are probably valid for batch sizes ranging from 128 to 512 and not only for a batch size of 256.

### 4.2. Experiment with Sub-Population Based on Sizes

#### 4.2.1. Datasets

We chose to rely on detection datasets for evaluating the capacity to predict distribution of sizes of objects of interest. We relied on the Munich Dataset for Vehicle Detection (MDVD) [31] and PASCAL VOC [32]. We tackled those detection datasets with the box proposal + rejection paradigm [33]. We acknowledge that the box proposal is clearly a deprecated paradigm for object detection, but it seems acceptable, as we only want to test the impact of different methods of estimating sub-population distributions.

For MDVD, we use 4K0G0060, 4K0G0070, 4K0G0080, 4K0G0090, and 4K0G0100 as the training set and 4K0G0010, 4K0G0020, 4K0G0030, 4K0G0040, and 4K0G0051 as the testing set. For PASCAL VOC, we sort the name of all images from the training set. Then, all images with IDs that are a multiple of three are used for testing and the other images are used for training (i.e., 66% training and 33% testing).

#### 4.2.2. Implementation Details

Specifically, we relied on selective search (opencv implementation [34]) for both MDVD and PASCAL VOC. Then, the classifier has only to decide if the box seems to be an object of interest. This classification is performed after resizing the image to 32 × 32. Yet, the true size (before resizing) is kept to regress the size distributions of accepted objects (as presented in Section 3.2.2) and to compare this predicted distribution to the true one. Indeed, the ultimate task is not to correctly accept/reject boxes but to ensure that the predicted size distribution matches the true one.

Importantly, as PASCAL is clearly larger than the other datasets, we performed the PASCAL experiment only once (instead using an average of five experiments).

Furthermore, *simple regression* is not presented, as it performs even worse than in class sub-population experiments. Dice is not a method by itself in these experiments as there is structurally more wrong boxes than correct boxes. Thus, dealing with the unbalance in the classification (for example, using Dice) is required for all methods.

Finally, in these experiments, we are focused on the size (in pixels) of the box’s diagonal. These sizes can be large (up to 300 for MDVD and 600 for PASCAL after removing outliers). Thus, the raw size distribution is sparse (given a single batch of data). However, using a simple blur as a kernel density function is sufficient to smooth the predicted/true density before comparison.

#### 4.2.3. Results

The results are presented in Table 5 and Table 6 for MDVD and PASCAL datasets.

The offered *density loss* is consistently better than the three baselines (i.e., *baseline with or without fairness or rejection*) on all backbones/datasets (except for RESNET34 on MDVD where *baseline+fairness* is 0.0015 higher than the offered loss). The consistent improvement brought by *density loss* can be explained by the fact that baselines are even less aware of the ultimate task than in the class sub-population case. Indeed, in both cases, a perfect classification still leads to a perfect distribution estimation. However, in the class case, the classifier outputs directly matches distributions, while in the size case, the distribution is extracted by combining classifier outputs and sizes that are unknown by the classifier. The offered paradigm is precisely a way to inject this information into the classification process, thanks to a careful implementation that allows a correct gradient flow.

Interestingly, best backbones for classification (such as RESNET) do not perform better for sub-population regression. This observation is already true for class-based sub-populations but is even clearer for size-based ones: despite a better accuracy, RESNET produces a worse size density estimate. Currently, this is partially but not only due to the small size of the boxes. It seems that there is also an impact of overconfidence (which can be harmful given the post-processing for extracting size density presented in Section 3.2.2). As a result, VGG13 is surprisingly the best backbone for size density estimation for the three baselines. This observation tends to prove that better classifiers do not imply better density regressors with standard loss. However, with the presented proportion loss, the performances of large backbones reach those of lighter backbones. This shows that making the classifier aware of the ultimate task may also allow for even larger architectures to be taken advantage.

## 5. Conclusions

This paper tackles the problem of estimating the distribution of different sub-populations within a batch of images. This problem has received almost no direct attention from the community, despite the fact that one can see the link with fairness, selective classification, multiple instance learning, and unstructured data classification (such as point cloud classification).

The main contribution of this paper is the demonstration, in five datasets, that baselines can be significantly improved by our approach based on the comparison of densities, allowing the intermediate classifier to be made aware of the ultimate task. The implementation of this loss (which allows an efficient propagation of the gradient within the comparison of densities) is carefully presented and the code is made available for the community.

In prospective work, we plan to combine our loss with the framework of Mask-RCNN to advance efficient size distribution estimation from object detection in images.

## Figures and Tables

**Figure 1 sensors-22-09218-f001:**
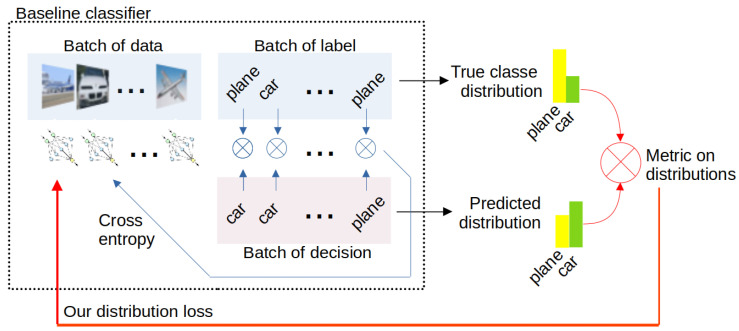
Overview of our approach for class-based sub-populations: our contributions are the consideration of the problem of class distribution estimation and the successful back-propagation of a metric on distributions into the baseline classifier.

**Figure 2 sensors-22-09218-f002:**
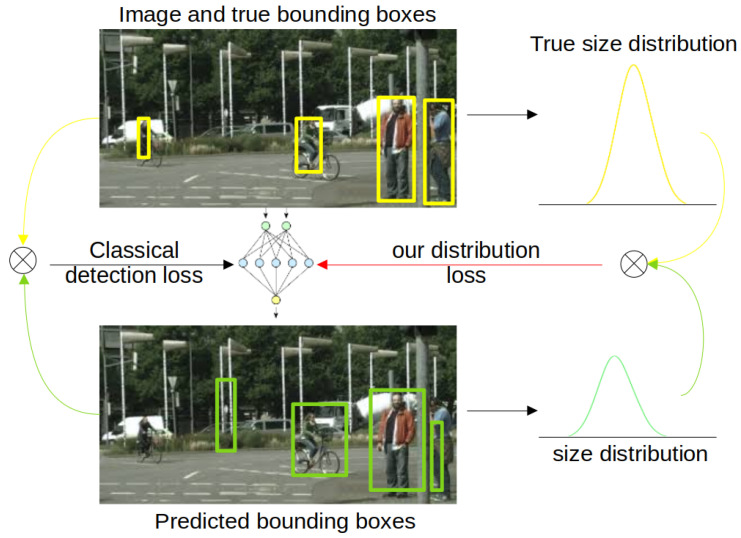
Overview of our approach for size-based sub-populations. Baseline detector is trained with classical loss (**left** of the figure). However, in this paper, we focus on extracting the size distribution of objects by efficiently back-propagating the metric on distributions into the deep network (**right** of the figure).

**Figure 3 sensors-22-09218-f003:**
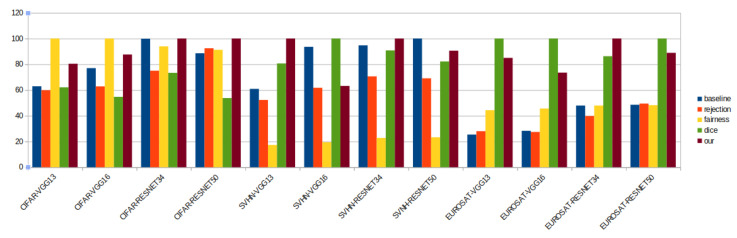
Relative performance of the different methods summarized in a graphic: one can see that the offered method is always either the best (100%) or nearly the best method, while other methods have worse average performance.

**Table 1 sensors-22-09218-t001:** Average distance (see Section 3.1) between true and predicted class distribution for different *methods* and backbones on **CIFAR10**. The two best results are in bold.

Methods	VGG13	VGG16	RESNET34	RESNET50
*simple regression*	0.2682	0.1626	0.2959	0.5024
*baseline*	0.1474	0.1191	**0.1146**	0.1329
*baseline+rejection*	0.1550	0.1460	0.1525	**0.1273**
*baseline+fairness*	**0.0927**	**0.0917**	0.1218	0.1290
*baseline+Dice*	0.1493	0.1677	0.1560	0.2190
*density loss* (ours)	**0.1153**	**0.1047**	**0.1144**	**0.1177**

**Table 2 sensors-22-09218-t002:** Average distance (see Section 3.1) between true and predicted class distribution for different *methods* and backbones on **SVNH**. The two best results are in bold.

Methods	VGG13	VGG16	RESNET34	RESNET50
*simple regression*	0.1983	0.1601	0.3115	0.8009
*baseline*	0.1053	**0.0775**	**0.0885**	**0.0880**
*baseline+rejection*	0.1226	0.1175	0.1187	0.1274
*baseline+fairness*	0.3707	0.3743	0.3682	0.3787
*baseline+Dice*	**0.0795**	**0.0725**	0.0923	0.1071
*density loss* (ours)	**0.0641**	0.1148	**0.0838**	**0.0972**

**Table 3 sensors-22-09218-t003:** Average distance (see Section 3.1) between true and predicted class distribution for different methods and backbones on **EUROSAT**. The two best results are in bold.

Methods	VGG13	VGG16	RESNET34	RESNET50
*simple regression*	0.3488	0.1562	0.4483	0.7276
*baseline*	0.2283	0.2189	0.1438	0.1368
*baseline+rejection*	0.2071	0.2259	0.1732	0.1343
*baseline+fairness*	0.1307	0.1358	0.1438	0.1378
*baseline+Dice*	**0.05792**	**0.0619**	**0.0799**	**0.0664**
*density loss* (ours)	**0.0682**	**0.0842**	**0.0689**	**0.0747**

**Table 4 sensors-22-09218-t004:** Influence of batch size on EUROSAT with RESNET50 backbone: we can see that results are relatively stable across batch sizes from 128 to 512.

Batch Size	128	256	512
*baseline*	0.1232	0.1212	0.1231
*density loss *(our)	0.0661	0.0420	0.0423

**Table 5 sensors-22-09218-t005:** Average distance (see Section 3.1 and Section 3.2.2) between true and predicted size distribution of objects of interest for different methods and backbones on **MDVD**.

Methods	VGG13	VGG16	RESNET34	RESNET50
*baseline*	0.2315	0.2503	0.2471	0.2390
*baseline+rejection*	0.2136	0.2303	0.2294	0.2262
*baseline+fairness*	0.1975	0.2112	**0.2098**	0.2130
*density loss* (our)	**0.1902**	**0.1961**	0.2113	**0.2107**

**Table 6 sensors-22-09218-t006:** Average distance (see Section 3.1 and Section 3.2.2) between true and predicted size distribution of objects of interest for different methods and backbones on **PASCAL VOC**.

Methods	VGG13	VGG16	RESNET34	RESNET50
*baseline*	0.4064	0.4256	0.4977	0.5192
*baseline+rejection*	0.4313	0.4594	0.5312	0.5500
*baseline+fairness*	0.4141	0.4319	0.4837	0.5070
*density loss* (our)	**0.4013**	**0.4201**	**0.4402**	**0.4268**

## Data Availability

All experiments from this paper are performed on public datasets and the code is publicly released.

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
