# Peer review of "Regressing Image Sub-Population Distributions with Deep Learning"

_sensors, 2022, doi:10.3390/s22239218_

Round 1
Reviewer 1 Report
(1) The problem to be solved in this paper is the parameter estimation problem of subclass distribution, which is a kind of classical problem in statistics and can be solved theoretically by the traditional sampling statistics method. However, this paper tries to learn the distribution from samples by machine learning method, and the estimated effect needs to be studied more carefully. First of all, statistical methods in classical statistics need a more comprehensive review; Secondly, the advantages and disadvantages of both need a more comprehensive comparison from theory to practice;
(2) The Introduction part of the paper is not clear about the background of the problem, which needs further combing;
(3) In Section 3, it is suggested that a diagram be provided to describe the working process of the proposed model;
(4) In Section 3, it is suggested that the influence of the sample size on the estimation performance is discussed;
(5) In Section 4, it is suggested that the proportion of training and test sets should be indicated;
(6) In Section 4, CIFAR10 data set is used in the experiment. The sample size of this data set is small, so it is suggested to introduce a larger data set for testing.
Author Response
First of all, I want to thank all reviewers for their comments.
We use red color for track change in the revised paper to make the second round more convenient for reviewers.
Addressing comments
The problem to be solved in this paper is the parameter estimation problem of subclass distribution, which is a kind of classical problem in statistics and can be solved theoretically by the traditional sampling statistics method. However, this paper tries to learn the distribution from samples by machine learning method, and the estimated effect needs to be studied more carefully. First of all, statistical methods in classical statistics need a more comprehensive review; Secondly, the advantages and disadvantages of both need a more comprehensive comparison from theory to practice;
We make it clearer that we are considering image batches and not just point batches. But this changes everything because because it requires to rely on deep networks due to image dimension, while those models are not suitable for classical statistical methods for subpopulation estimation.
So we precise in all the paper (title, abstract, introduction, related works, methodology, experiment and conclusion) that we are focusing on image data.
The Introduction part of the paper is not clear about the background of the problem, which needs further combing;
Currently, the targeted application is aluminium particle size distribution estimation during propellant combustion (as fragmentation and agglomeration change the distribution of the injected particles). However, due to the sensibility of those data, we can not offer experiments on them at this point. However, we have added a sentence to precise this use case.
In Section 3, it is suggested that a diagram be provided to describe the working process of the proposed model;
We thank reviewers for this advice which makes the paper much clearer. This has been added in the revised version.
In Section 3, it is suggested that the influence of the sample size on the estimation performance is discussed;
There are two "sizes" that have influence: batch size (from which we extract distribution) and image size for when regressing size distribution.
For image size, it is known that objects with bounding boxes smaller than 32x32 are harder to detect. But this is exactly the purpose of our paper to improve the final distribution by asking the network to try to be as good for small objects as for large objects (potentially just worsening detection for large ones but still improving distribution at the end).
For batch size, it is clear that this parameter has strong influence: if batch size is 1, then the targeted problem collapses in classification, if batch size is very large, then distribution matches the global statistic at dataset level. So, the problem is interesting only for middle batch size.
So, we thank the reviewer for this advice of considering the sensibility to this batch size. We have added a table with results with batch size going from 128 to 512 and a paragraph to clarify this point.
In Section 4, it is suggested that the proportion of training and test sets should be indicated;
We thank the reviewer for this advice. This has been added in the revised version.
In Section 4, the CIFAR10 dataset is used in the experiment. The sample size of this data set is small, so it is suggested to introduce a larger data set for testing.
Currently, we believe that the offered framework is interesting exactly in the context of a small dataset. Indeed, estimating size distribution from a batch is relevant when the test set is small. Otherwise one could average information across the full dataset and somehow recover the precise distribution.
This is why we target only moderate datasets like CIFAR10 or PASCAL VOC. Also, those datasets are quite classical for the computer vision community allowing the audience to have a better intuition of the content (despite our problem being weakly considered today).
Reviewer 2 Report
In this study, deep learning based regressing sub set distribution were presented. Topic is interesting but some revisions should be done. The abstract is insufficiently informative. Comments are given below:
1. The abstract is insufficiently informative. Your achievements should be given using your results. Trained model are not given in the abstract and too little novelty given in this section. State of the art of the study is not clearly defined in the paper.
2. Literature search given in a mass. Reading this introduction and understanding the literature study are too hard for readers. Please give your distribution (for regression) in experiments.
5. Methods should be compared by using success rates for similar studies in a separate section like “Discussion”.
6. There is no block schematic in the paper. Please clearly define in the paper.
7. Your future aspects should be given in details in Conclusion.
Author Response
First of all, I want to thank all reviewers for their comments.
We use red color for track change in the paper to make the second round more convenient for reviewers.
Addressing comments
The abstract is insufficiently informative. Your achievements should be given using your results. Trained model are not given in the abstract and too little novelty given in this section.
We add precise results in the abstract both in terms of performance or in terms of backbone.
State of the art of the study is not clearly defined in the paper. Literature search given in a mass. Reading this introduction and understanding the literature study are too hard for readers.
As pointed out by both reviewers, the state of the art section presents some concepts related to the paper (e.g. fairness) rather than clear comparable works. However, to our knowledge, we are the first to try to deal with size or class distribution estimation from a batch of images data. Indeed, this problem is hard because it requires to rely on deep networks due to image dimension, while those models are not suitable for classical statistical methods for subpopulation estimation.
Please give your distribution (for regression) in experiments.
Currently, the purpose of the paper is to deal with real image dataset. So we do not offer toy experiments where distribution would be known (and then compared to the regressed ones). In addition, such low dimensional regression experiments would likely not require deep networks: using statistical ways would be more convenient. Inversely, extracting the correct size distribution while detecting objects on PASCAL VOC is definitely challenging.
To emphasize this point, we add a sentence in the revised version pointing to the fact that small objects are known to be harder to detect e.g. objects with bounding boxes smaller than 32x32. Yet, our paper wants to improve the final distribution by asking the network to try to be as good for small as for large (potentially just worsening detection for large but still improving distribution at the end).
Methods should be compared by using success rates for similar studies in a separate section like “Discussion”.
Indeed, we acknowledge that our experiments are rather ablation study than real comparison with SOTA. But, to our knowledge there is no comparable study. The most comparable is maybe "pointnet" (called "direct regression" in the paper) which is designed for 3D point batches processing while we tackle image batches. This explains why this method based on a pooling to deal with a batch of data fails in our cases.
There is no block schematic in the paper. Please clearly define in the paper.
We thank reviewers for this advice which makes the paper much clearer. This has been added in the revised version.
Your future aspects should be given in details in Conclusion.
We are currently working on the use of our loss with mask-rcnn to improve size distribution estimation in detection. This point has been added to the conclusion.
Round 2
Reviewer 1 Report
The author has carefully modified the suggestions and the quality of the paper has been improved. It is recommended to be accepted.
Reviewer 2 Report
All revisions have been made, except for the separate discussion section, but this version is quite adequate.